# A Novel Benchtop Device for Efficient and Simple Purification of Cytokines, Growth Factors and Stem Cells from Adipose Tissue

**DOI:** 10.3390/biomedicines11041006

**Published:** 2023-03-24

**Authors:** Martina Semenzato, Ludovica Zambello, Stefania Fumarola, Enrico Motta, Luana Piroli, Luca Scorrano, Camilla Bean

**Affiliations:** 1Department of Biology, University of Padova, Via U.Bassi 58/B, 35121 Padova, Italy; 2Veneto Institute of Molecular Medicine, 35129 Padova, Italy; 3InScientiaFides Foundation, Strada di Paderna, 2, 47895 San Marino, San Marino; 4Motta e Partners Stp Srl, 25122 Brescia, Italy; 5Department of Medicine, University of Udine, Piazzale Kolbe 4, 33100 Udine, Italy

**Keywords:** mesenchymal stem/stromal cells, regenerative medicine, ADSCs, adipose tissue, immunomodulatory cytokines, mechanical dissociation, lipoaspirates

## Abstract

Lipoaspirates represent a source of adult stem cells, cytokines, and growth factors of adipocyte origin with immunomodulation and regenerative medicine potential. However, rapid and simple protocols for their purification using self-contained devices that can be deployed at the points of care are lacking. Here, we characterize and benchmark a straightforward mechanical dissociation procedure to collect mesenchymal stem cells (MSCs) and soluble fractions from lipoaspirates. IStemRewind, a benchtop self-contained cell purification device, allowed a one-procedure purification of cells and soluble material from lipoaspirates with minimal manipulation. The recovered cellular fraction contained CD73^+^, CD90^+^, CD105^+^, CD10^+^ and CD13^+^ MSCs. These markers were comparably expressed on MSCs isolated using IstemRewind or classic enzymatic dissociation procedures, apart from CD73^+^ MSCs, which were even more abundant in IStemRewind isolates. IstemRewind-purified MSCs retained viability and differentiation into adipocytes and osteocytes, even after a freezing-thawing cycle. Levels of IL4, IL10, bFGF and VEGF were higher compared to the pro-inflammatory cytokines TNFα, IL1β and IL6 in the IStemRewind-isolated liquid fraction. In sum, IStemRewind can be useful for straightforward, rapid, and efficient isolation of MSCs and immunomodulatory soluble factors from lipoaspirates, opening the possibility to directly isolate and employ them at the point-of-care.

## 1. Introduction

Adipose tissue is regarded as an excellent source of mesenchymal stem cells (MSCs) called adipose-derived stem cells (ADSCs) for plastic surgery and regenerative medicine. Cells can be harvested through liposuction, a very common relatively safe and simple cosmetic procedure that allows easy access to a large amount of adipose tissue. Compared to other tissues and organs used as source of MSCs, including bone marrow, peripheral blood, the umbilical cord, placenta, amniotic fluid, and dental pulp, fat depots easily yield high amounts of MSCs [1,2,3,4,5]. ADSCs display a high proliferation rate and differentiation multipotency. Moreover, they secrete multiple cytokines and proteins potentially associated with immunoregulatory and anti-inflammatory effects, making them an ideal biomedicine for tissue repair [6]. Currently, there are more than one thousand registered MSC clinical trials [7]. In general, MSCs obtained from adipose tissue are useful for several applications in regenerative therapy. They have been used to treat autoimmune disease [8] and inflammatory diseases [9] including osteoarthritis and rheumatoid arthritis [10,11], and to promote regeneration [12]. ADSCs therapy is used also to treat ischemic heart disease, one of the leading causes of premature death worldwide [13,14]. Although autologous cells are of course less immunogenic, their ability to favor tumorigenesis remains a serious safety concern [15,16,17], prompting the development of protocols to also isolate from lipoaspirates cell-free extracts that can be enriched in growth factors and immunomodulatory cytokines. Furthermore, a hurdle in the use of MSCs in clinical practice is represented by the lack of standardized, reproducible preparation methods that can be deployed at the point of care for autologous ADSCs’ isolation and immediate administration to the patient.

ADSCs can be obtained from the stromal vascular fraction (SVF), also composed of blood, endothelial cells, and fibroblasts, by enzymatic or mechanical dissociation of adipose tissue. Enzymatic dissociation based on collagenase digestion is the most efficient method and is therefore considered the gold standard. However, this procedure is difficult to standardize because collagenase activity may vary across batches and manufacturers [18]. Moreover, this method is expensive, time-consuming, and requires manipulation by personnel with laboratory experience as well as at least a laminar flow hood to maintain the sterility of the isolated MSCs [19]. Finally, enzymatic digestion may alter the biological properties of ADSCs. Not surprisingly, several alternative non-enzymatic methods that are cheaper, faster, and simpler have therefore been developed to obtain ADSCs [20,21,22]. Here, we benchmarked a very quick non-enzymatic protocol (patent IT201700102994A1) that is self-contained in the benchtop medical device IStemRewind (GenLife) against gold-standard enzymatic dissociation to obtain ADSCs and soluble factors from human lipoaspirates donated by individuals undergoing cosmetic liposuction. Our results indicate that IStemRewind isolates comprise MSCs and a soluble fraction containing mostly growth factors and immunomodulatory cytokines rather than pro-inflammatory cytokines. Both the cellular and the soluble fractions are qualitatively comparable to those obtained after enzymatic dissociation of adipose tissue by skilled personnel in the laboratory. The IStemRewind-isolated MSCs can be cryopreserved in the long term without affecting their properties of self-renewal and multipotency in vitro, thus potentially allowing their biobanking for delayed personalized medicine applications.

## 2. Materials and Methods

### 2.1. Materials and Reagents

The cell culture medium was Dulbecco’s modified eagle medium with 4.5 g/L glucose (DMEM, Merk Life Science, Milano, Italy), 10% decomplemented fetal bovine serum (FBS, Life Technologies, Monza, Italy), and 1% penicillin/streptomycin (P/S, Life Technologies, Monza, Italy). Dulbecco’s phosphate-buffered saline (DPBS) and trypsin-EDTA were purchased from Sigma. Dimethyl sulfoxide (DMSO) was purchased from Life Technologies. Cell culture reaction tubes and cell culture plates and flasks were purchased from Corning Inc. (Corning, NY, USA). The freezing containers used for cryopreservation were Nalgene purchased from Merk Life Science.

### 2.2. Lipoaspirates Collection

Adipose tissue from 17 individuals (age range: 24–61 years; mean age: 44.5 ± 9.7) who underwent aesthetic liposuction was used. All individuals provided informed consent. All procedures were carried out in accordance with the declaration of Helsinki in its latest amendment. Fat harvesting involved standard procedures of lipoaspiration from subcutaneous depots through use of a cannula. The subcutaneous layer was infiltrated with large volumes of crystalloid fluid (Klein solution) containing a local anesthetic agent (lignocaine, epinephrine, and saline), thereby eliminating the need for general anesthesia and limiting blood loss. Fat tissue was then aspirated and transferred into a 50 mL syringe.

### 2.3. Mesenchymal Cells Extraction from Adipose Tissue Using IStemRewind

The IStemRewind system (GenLife) is a biological extractor equipped with a sterile disposable kit. The processing technique is enzyme-free, and the final homogenate is an immunoregulatory factor-enriched solution which contains MSCs. A closed circuit guarantees sterility, allowing the utilization of the biological material in a straightforward way after the process at the point of care. The medical device IStemRewind applies a patented method (IT201700102994A1) of active factors’ extraction in a short time. This benchtop simple device was conceived for use by any minimally trained surgeon for the mechanical dissociation of adipose tissue (Figure 1a). The biological material is processed in a sterile disposable tube (Figure 1b). The static conveyors, located near the bottom of the container, are suitably inclined and designed to convey the biological material against the walls of the tube while the blade is rotating. The dynamic flow generated causes a pressure that efficiently homogenizes the adipose tissue. The separation device allows a selective and rapid extraction of the active regenerative biomedicines, compatible with the use in the operating room and within the operating time. Some 20 mL of adipose tissue from liposuctions contained in 50 mL sterile syringes was directly injected into the IStemRewind tube using port A (Figure 1). Using the same injection point, 20 mL of sterile saline solution was added. Port A was then sealed, and the IStemRewind tube was connected to the IStemRewind machine and processed using the following program: 3000 rpm for 50 s, hold for 10 s and 600 rpm for 20 s. After completion of the homogenization program, the IStemRewind tube was held upside down using the IStemRewind stand for 15 min. The liquid phase and the cellular pellet were collected using a sterile 50 mL syringe connected through port B of the IStemRewind tube (Figure 1). If required, the homogenate was centrifugated at 1500 rpm for 10 min at room temperature (RT) to separate cells from the liquid phase.

### 2.4. Mesenchymal Cells Extraction Using Collagenase

Mesenchymal cells were isolated from lipoaspirates using the procedure described in [23]. Adipose tissue was transferred from the lipoaspirates syringe to a sterile 10 cm round Petri dish. After mincing with sterile scissors, the tissue was transferred to a 50 mL tube and incubated with 0.2% (*w/v*) collagenase type I in DMEM for 1 h at 37 °C in a shaking water bath. The digested tissue was transferred to a 10 mL tube and centrifuged at 322× *g* for 10 min. The pellet was resuspended in DMEM (Gibco) supplemented with 3% FBS, filtered through a sterile strain and centrifuged again at 322× *g* for 10 min. The pellet was resuspended in 5 mL of RBC lysis buffer (Thermo Fisher Scientific, Rodano, Italy) for 5 min. To eliminate fat residues, the solution was transferred to a new 10 mL tube and further centrifuged at 322× *g* for 10 min. The pellet was resuspended in 10 mL of DMEM F12 with 3%FBS and filtered through a 100 mm cell strainer (Falcon). The collected cells were counted and grown in DMEM at 37 °C and 5% CO_2_ in a water-jacketed incubator.

### 2.5. Cell Seeding and Expansion

After reaching 80–90% confluence, cells were passaged into new dishes or flasks. The supernatant was aspirated, and cells were washed with DPBS to remove residual FBS contained in the cell medium. Trypsin/EDTA solution was added and incubated over the cell surface until cells started to detach from the plastic support. To halt the enzymatic activity, a proper amount of complete DMEM was added. The cell suspension was then collected into a new reaction tube and centrifuged at 500× *g* for 10 min. The pellet was resuspended in warm medium and seeded in a new, larger cell culture flask. Cells were maintained at 37 °C and 5% CO_2_ in a water-jacketed incubator.

### 2.6. Freezing and Thawing

Cells were detached from the cell culture plate or flask using trypsin-EDTA. After cell counting, cells were centrifugated and the supernatant was aspired. 1 × 10^6^ cells were then resuspended in 1 mL of freezing medium constituted of fetal bovine serum containing 7% DMSO. The cell suspension was transferred into freezing tubes and put into a freezing container with isopropanol (Mr. Frosty, Nalgene), which provides a consistent freezing rate of 1 °C/min when stored at −80 °C. After 1 day at −80 °C, the tubes were transferred in liquid nitrogen for long-term storage. When required, cells were thawed by placing the plastic vial on a 37 °C water bath. Cells were then transferred into a 15 mL reaction tube and supplemented with 10 mL of complete cell culture warm medium at 37 °C. After centrifugation at 300× *g* for 10 min, the supernatant was aspired, and cells were resuspended in fresh complete DMEM and seeded in a new 10 cm round tissue culture dish.

### 2.7. Flow Cytometry

Cells from different donors were used to characterize their immunoprofile. This was performed in isolated cells after reaching confluence in a 10 cm plastic dish (Corning), usually between passage 9/10 and in cells after freezing/unfreezing. After trypsinization, cells were counted and collected in a cytofluorimetric tube. Cells (1 × 10^5^ for each condition) were centrifugated and resuspended in 100 µL of DPBS. Upon resuspension, cells were left unstained (unlabeled control) or incubated with the indicated antibodies for 10 min at RT in the dark. The commercially available antibodies FITC-CD19, PeCy5.5-CD14, APC-CD34, PE-CD73, APC-HC-CD45, PE-HLA-DR, FITC-CD90, PeCy5.5-CD105, PE-CD10, FITC-CD36, APC-CD13 and the isotypic controls FITC-IgG1, PE-IgG1, PeCy5.5-IgG1 were purchased from Becton Dickinson (Sunnyvale, CA, USA). After incubation, cells were washed in DPBS and finally resuspend in 100 µL of DPBS. Cells were analyzed using FACSCanto analyzer (BD Biosciences, San Jose, CA, USA) and processed using by the BD FACSDiva software program (BD Biosciences).

### 2.8. Quantification of Growth Factors Content

Growth factor content was quantified in the total fraction and in the liquid phase of adipose tissue sample, and protein content-normalized values were depicted in the form of a boxplot. Samples were prepared following the manufacturer’s instruction. Cytokines and growth factors were quantified in the supernatants using specific ELISA kits for the indicated cytokines (R&D Systems).

### 2.9. ADSCs Differentiation Assays

In vitro differentiation into adipocytes and osteocytes was performed on MSCs at passage P3/4. For the adipocyte differentiation assay, 5.0 × 10^4^ cells/well were seeded in a 6-well plates. Once at confluence, cells were fed with an adipogenic induction medium (MAD), composed of DMEM/F12 containing 5% FBS, 100 nM dexamethasone (Sigma-Aldrich, St. Louis, MO, USA), 1 nM T_3_ (Sigma-Aldrich), 66 nM insulin (Sigma-Aldrich), 17 µM pantothenate (Sigma-Aldrich), 33 µM biotin (Sigma-Aldrich) and 10 µg/mL transferrin (Sigma-Aldrich) supplemented with 0.25 mM IBMX (Sigma-Aldrich), 10 µM rosiglitazone (Sigma-Aldrich) for three days, as described in [23]. Differentiated cells were maintained in MAD medium.

Adipocyte differentiated cells were stained with Oil Red O (Sigma-Aldrich). Cells were plated on a 6-well plate, were washed in DPBS and fixed in 3.7% formalin (Sigma-Aldrich) for 15 min. After washing in DPBS, cells were stained with freshly prepared Oil Red O solution for 30 min at RT. Cells were washed 3 times in MilliQ water and imaged.

For osteogenic differentiation, 5.0 × 10^4^ cells/well were seeded in a 6-well plate. After reaching confluence, cells were provided with an osteogenic medium consisting of DMEM supplemented with 10% FBS, 10 mM β-glycerophosphate (Sigma-Aldrich), 50 µM ascorbate-2-phosphate (Sigma-Aldrich), 100 nM dexamethasone (Sigma-Aldrich), 1% P/S (Sigma-Aldrich). After 20 days of incubation, cells were stained with Alzarin Red S (Sigma-Aldrich) to assess the degree of mineralization. Briefly, the medium was removed, and the cells were washed twice with DPBS and then fixed with 95% methanol for 10 min. Cells were washed in DPBS and then incubated for 45 min with 2% Alzarin Red S working solution in the dark. The dye was removed, and cells washed twice with DPBS and imaged.

For bright field imaging of ADSC and differentiated cells, tissue culture dishes were placed on the stage of an inverted Olympus microscope (CKX53) equipped with a LCACHN20XIPC, 0.4 NA, 3.2 WD objective and a DP23 digital camera.

### 2.10. RNA Extraction and qRT-PCR

Total RNA was extracted from primary cells using Trizol (Thermo-Fisher Scientific). RNA content was quantified using Nanodrop One (Thermo-Fisher Scientific) and purity was assessed by absorbance ratios of 260:280 nm and 260:230 nm. From an equal amount of total RNA of each sample, cDNA was generated using an iScript cDNA synthesis kit (Biorad Laboratories, Segrate, Italy). Quantitative real-time PCR (qRT-PCR) was carried out with SYBR Green chemistry (Promega, Milano, Italy) using the Quant Studio 5 Real-Time PCR System (Thermo-Fisher). Primer sequences were the following:LEPTINFw: GTGCGGATTCTTGTGGCTTTRv: GGAATGAAGTCCAAACCGGTGFABP4Fw: GCCAGGAATTTGACGAAGTCACRv: TTCTGCACATGTACCAGGACACRUNX2Fw: CTCCGGCCCACAAATCTCRv: CACGACAACCGCACCATBGLAP (OSTEOCALCIN)Fw: AGCAAAGGTGCAGCCTTTGTRv: GCGCCTGGGTCTTCACTβ-ACTINFw: AGAGCTACGACTGCCTGACRv: GGATGCCACAGGACTCCA.

Samples (10 ng of cDNA) were normalized by β-ACTIN content and reported as a fold increase or decrease vs. the control.

### 2.11. Statistical Analysis

Data are presented in box/dot plot graphs. Dots represent the individual measurement, boxes represent the mean ±s.e.m, and whiskers represent the 10th and 90th percentiles. Normal distribution of data in each sample with individual measurement was verified by a Shapiro–Wilk test. If data were not normally distributed, a non-parametric Kruskal–Wallis ANOVA was used to analyze significance. A *p* ≤ 0.05 was considered significant. Graphs and statistical analyses were performed using OriginPro, Version 2022 (OriginLab Corporation, Northampton, MA, USA).

## 3. Results

### 3.1. Lipoaspirate Processing Using IStemRewind

Non-enzymatic systems for obtaining ADSCs from adipose tissue can represent a safe and effective tool in regenerative medicine. We therefore set out to analyze the performance of IstemRewind (GenLife, Figure 1). A volume of 20 mL of lipoaspirate samples, delivered in sterile syringes directly from the clinic (Figure 2a), was directly injected into a sterile IStemRewind tube through the sealable entrance port (Figure 2b). After adding 20 mL of sterile physiological solution to dilute the adipose tissue, the IStemRewind tube (ISR tube) was connected to the IStemRewind device and processed as described in the Methods section (Figure 2c,d). At the end of the procedure, the ISR tube was placed upside down using the holder of the device to allow sedimentation. After the sedimentation, four phases could be detected, from the top: oil, condensed fat, a liquid acellular phase, and a cellular fraction (Figure 2e,f). Using a Luer lock syringe, the liquid phase and the cellular fractions (total fraction, TF) were collected and transferred in a sterile tube. A volume of 5 mL of TF was immediately snap-frozen in liquid nitrogen for subsequent analyses (Figure 2g). The remaining homogenate was centrifuged to isolate the vascular stromal fraction which contains mesenchymal stem cells (MSCs) as well as blood, endothelial cells, and fibroblasts. The acellular liquid phase (LP) was immediately snap-frozen in liquid nitrogen for subsequent analyses (Figure 2h).

### 3.2. Lipoaspirate-Derived Cells Can Be Expanded in Culture

We first cultured and morphologically analyzed the cell fraction from the IStemRewind extraction from the aspirate samples. Freshly isolated cells were properly expanding, and when reaching a confluence of 80–90%, they were easily subcultured into new flasks upon trypsinization. Cells were imaged to observe their morphology at different time points. The morphology of growing cells showed typical MSCs features, with fibroblastic-like morphology (Figure 2i–k).

### 3.3. Lipoaspirate-Derived Isolates Contain Cells with MSCs Markers

Cell type identification in complex multicellular suspensions can be achieved by flow cytometric profiling of the surface expression of the cluster of differentiation (CD) markers. We therefore analyzed the cells harvested using IStemRewind for the presence of typical MSCs markers by flow cytometry [24]. We profiled the isolated cells for MSCs markers (CD73, CD90, CD105, CD10 and CD13) and markers for hematopoietic lineages (HLA-DR, CD45, CD19, CD14, CD34 and CD36). Cells positive for the different MSCs markers represented 60–100% of the cells obtained using the IStemRewind procedure, while cells positive for hematopoietic markers represented, at most, 3% of the total cellular population (Figure 3a–c). CD34^+^ cells were more frequent than other “non-MSCs” cells. However, CD34 is not only a marker of hematopoietic precursors, but also of MSCs and of other non-hematopoietic precursors [25], potentially explaining why the CD34^+^ population appeared relatively enriched in the IStemRewind isolates. Thus, immunophenotyping indicates that IStemRewind isolates are enriched in cells with MSCs surface profiles.

### 3.4. Cells Isolated from Lipoaspirates Can Be Differentiated into Adipocytes and Osteocytes

While surface markers analysis indicated that the cells isolated from lipoaspirates using IStemRewind displayed MSCs features, these experiments did not provide evidence on their differentiation potential. To this end, we decided to verify whether these cells retained features of multipotency by culturing them under conditions that induce targeted commitment, including to osteogenic and adipogenic lineages (Figure 4a–f). Ten days after adipogenic differentiation induction, cells from each individual fat sample contained several lipid droplets that were stained using specific Red Oil O staining (Figure 4b,e). When we induced osteogenic differentiation, we could appreciate not only the typical morphological changes, but also the deposition of calcium precipitates that were positive for the specific Alizarin Red staining (Figure 4c,f). In addition to these histological analyses, we wished to profile the expression of specific genes that characterize the two studied lineages. qRT-PCR of retrotranscribed mRNA from ADSCs from six different individuals indicated the consistent induction of the specific adipocyte *LEPTIN* and *FABP* when cells were cultured for 4 days in adipogenic media, and of the osteocyte *RUNX2* and *OSTEOCALCIN* markers when ADSCs were kept in osteogenic differentiation media for 10 days (Figure 4g,h). Altogether, these experiments indicate that IStemRewind can efficiently purify ADSCs that retain the differentiation potential typical of MSCs derived from fat.

### 3.5. IStemRewind Isolates Are Enriched in CD73^+^ MSCs Compared to MSCs from Enzymatic Dissociation Procedure

After having established that IStemRewind allows the isolation from lipoaspirates of cells that can be kept in culture and display surface markers, as well as the differentiation potential typical of ADSCs, we wished to benchmark this procedure with the collagenase digestion that is considered the gold standard for the isolation of vascular stromal cells. We obtained proliferating cells from lipoaspirates by enzymatic dissociation of adipose tissue or by using IStemRewind, and we grew them at confluency (Figure 5a,b). Flow cytometric counting of the cells isolated using the two procedures indicated a higher yield for the enzymatic method (Figure 5c). We next compared the expression of different surface markers by flow cytometry in cells isolated using IStemRewind and kept in culture, both in cells freshly extracted from lipoaspirates using collagenase digestion and in cells from lipoaspirates extracted using the collagenase method and kept in culture for the same number of passages used for the IStemRewind isolated cells. Of note, irrespective of the method used to isolate the cells, positivity levels for the typical MSCs surface markers CD90 and CD105 were comparably high, whereas cells positive for the hematopoietic markers HLA-DR, CD45, CD19 and CD14 were rare (Figure 5d). Interestingly, loss of surface markers is a known side effect of MSCs isolation using enzymatic collagenase digestion [26]. Indeed, approx. 50% of the collagenase isolated MSCs were CD73^−^ already at the time of isolation, whereas we found that most of the IStemRewind freshly isolated cells retained expression of this marker, the expression of which has been associated with a cell population that can favor cardiac repair and delay the progression of rheumatoid arthritis [27,28]. Moreover, IStemRewind isolates were enriched in cells positive for CD34 compared to collagenase isolated cells kept in culture for the same number of passages. While CD34 has been traditionally regarded as a hematopoietic stem cell marker, it is now considered a typical MSCs marker that identifies a subset of cells with enhanced differentiation ability [25]. Altogether, these data indicate that while the yield of cells obtained using IStemRewind is lower than that of classic enzymatic methods, this procedure enriches for ADSCs positive for CD73 and CD34.

### 3.6. The Total Fraction and Liquid Phase Isolated Using IStemRewind Contain Anti-Inflammatory Cytokines and Growth Factors

While the use of MSCs offers great promises in regenerative medicine, safety concerns due to their tumorigenic potential exist. Moreover, their use is limited to autologous cells to limit of course adverse immunological reactions. Conversely, adipose tissue also represents a source of several soluble factors (cytokines, growth factors) with potential immunomodulatory features that are devoid of the immunological and the long-term safety concerns of MSCs. We therefore decided to explore whether cytokines and growth factors were retrieved in the cellular as well as in the soluble fractions extracted from lipoaspirates using IStemRewind. To this end, we first measured by specific ELISA the amount of immunomodulating as well as proinflammatory soluble factors in the total fraction (TF) of the IStemRewind isolates from 17 lipoaspirates. Interestingly, we found that in the TFs, the concentration of the growth factors and immunomodulatory cytokines IL4, IL10, bFGF and VEGF were higher compared to the concentration of the pro-inflammatory cytokines TNFα, IL1β and IL6 (Figure 6a). However, TF also contains the cellular component, and its potential clinical use as a source of cytokines and growth factors is affected by the same safety concerns highlighted for the ASDCs. We therefore decided to compare the levels of these soluble factors in the TF versus the cell-free liquid phase (LP) obtained from the centrifugation of the TF of the IStemRewind isolates. In a subset of six analyzed samples, we found that levels of the measured cytokines and growth factors were comparable in TFs and LPs (Figure 6b). Thus, LPs obtained by a single centrifugation step from the TFs isolated using IStemRewind from lipoaspirates contain immunomodulatory cytokines and growth factors that may also represent nonimmunogenic biomedicines for potential allogeneic use.

### 3.7. Adipose Tissue Derived Cells Can Be Cryopreserved

One of the potential advantages of ADSCs isolated from lipoaspirates is that they can be potentially banked to provide a source of MSCs that can be expanded for future use. To test whether cells obtained from lipoaspirates using IStemRewind retained their growth potential after cryopreservation, we rapidly snap-froze a small amount (5 mL) of the resuspended pellet containing the vascular stromal fraction from the IStemRewind isolates. This cellular pellet was suspended in cryopreservation media, frozen to −80 °C at a controlled rate, transferred to the vapor phase of a liquid nitrogen cryotank and stored for one month. We then thawed the material and verified whether cells were retrieved and could grow in culture. Not only we could grow these cells, but we could also expand and differentiate them into adipocytes and osteocytes (Figure 7b,d,f,h) to a degree that was comparable to those obtained from the freshly isolated counterparts (Figure 7a,c,e,g). Thus, IStemRewind isolates contain ADSCs that can be cryopreserved without losing their ability to differentiate.

## 4. Discussion

Several clinical and preclinical trials have indicated the potential of ADSCs to treat different diseases [29]. However, their use in the clinics is limited by the length and need of specialized equipment and trained personnel at the point of care to isolate them from lipoaspirates. Here, we show that IStemRewind, a simple, easy to use benchtop instrument that does not require a sterile flow hood or trained personnel can efficiently isolate CD73 and CD34 positive ASDCs as well as in immunomodulatory cytokines and growth factors from small amounts of lipoaspirates.

The attractiveness of ADSCs in regenerative medicine arises from several of their characteristics; they are abundant and can be retrieved at a site that is accessible through a minimally invasive and ethical procedure such as lipoaspiration. They are multipotent, being able to differentiate into multiple cell types including adipocytes, osteoblasts and chondrocytes; they display a low level of immunogenicity [30] and possess immunomodulatory properties through autocrine and paracrine factors [31]. Human ADSCs are mainly isolated from lipoaspirates containing aggregates of adipocytes, collagen fibers, and the stromal vascular fraction (SVF) that is enriched in different cell types, including ADSCs, mesenchymal and endothelial progenitor cells, leukocytes, lymphatic cells, pericytes, and vascular smooth muscle cells.

Mechanical or enzymatic methods have been traditionally used to isolate ADSCs-containing fractions from lipoaspirates. Although enzymatic digestion by collagenase is the gold-standard method, it is time-consuming and requires invasive manipulations of the sample which might affect the sterility and integrity of cells and require specialized personnel and instrumentation, often not available at the point of care. For these reasons, a continuous effort is being made to develop new techniques which aim for a closed, sterile, and safe isolation process, limiting risk for contaminations [32]. Indeed, several manual and automated systems are now commercially available for the isolation of SVF directly in the operating room at the point of care. These methods cannot be compared in terms of process time, volume capacity, yield, viability, surface marker identity, safety profile of the cells, and capital and operating costs. Moreover, they show a significant variability in the number, identity, and safety profiles of the viable recovered cells.

We analyzed the performance of a newly developed mechanical system that can be operated by a single trained individual at the point of care. We found that vis à vis a very rapid processing time (ranging from 15 to 30 min), IStemRewind achieved a complete extraction of the vascular stromal fraction containing the ADSCs and of the liquid phase enriched in immunomodulatory cytokines and growth factors. These ADSCs not only consistently expressed known mesenchymal stem cell markers including CD73, CD90, and CD105; they were only marginally contaminated by hematopoietic cells marked by CD14, CD19, CD45, and HLA-DR. CD34^+^ cells were a major component of the IStemRewind isolates. CD34 has been traditionally considered a marker of hematopoietic cells, but it is now considered a hallmark of a subset of MSCs with higher regenerative potential [25]. Because its expression on MSCs is lost during their in vitro expansion [33,34,35], an enrichment in CD34 positive cells might represent an advantage if ADSCs from lipoaspirates are destined for cryopreservation and biobanking. Notably, several assays demonstrated that the ASDCs isolated using the IStemRewind system maintained their capacity to differentiate into adipocytes and osteoblasts, even after one year of cryopreservation (not shown), indicating that the IStemRewind isolation does not damage the purified cells.

To benchmark the performance of IStemRewind for isolation of ADSCs from lipoaspirates, we compared it with the standard enzymatic method based on collagenase digestion. While a higher number of cells were obtained by collagenase digestion, the cells isolated mechanically by IStemRewind showed comparable multipotency in differentiation assays. Cells isolated using IStemRewind were consistently CD73^+^CD90^+^CD105^+^, whereas in the digestion isolates, we found a proportion of CD90^+^CD105^+^ cells that were also CD73^−^. Indeed, the use of collagenase for MSCs isolation has been reported to impact the levels of MSCs surface markers [26]. These results suggest that IStemRewind might conversely preserve surface markers expression, opening further clinical applications for IStemRewind isolates. For example, increased expression of CD73, a key enzyme involved in adenosine production, is associated with reduced macrophage infiltration and inhibited fibrosis development [36]. Moreover, given their anti-inflammatory activity, CD73^+^ MSCs can favor cardiac repair [27]. It is tempting to speculate that IStemRewind-obtained ADSDs could be used in the prevention of the consequences of myocardial infarction, even at the point of care during percutaneous catheterism and reperfusion.

We characterized the aqueous phase of the IStemRewind isolates from lipoaspirates. This fraction contained key immunomodulatory factors. As the paracrine effects of this aqueous phase may be more therapeutically promising than the ability of ADSCs to regenerate tissues [37], the extracted fraction containing multiple cytokines, chemokines, and growth factors might represent another biomedicine for regenerative medicine. Importantly, the liquid and the cell fractions from the obtained aqueous phase can be easily separated by a simple centrifugation step, and both products can be easily stored for a long time by cryopreservation. Cryopreserved cells from the IStemRewind isolates maintained their capacity for self-renewal and differentiation, presenting the possibility to create a biobank of ASDCs. Such a biobank might represent a therapeutic advantage, as the secretory capacity of the adipose tissue and the ACDCs’ yield decrease upon ageing [38,39]. In this perspective, ADSCs could be stored for a potential future use in autologous transplantation, and the immunomodulatory liquid phase could be used in autologous or allogenic treatment. Indeed, this cell-free phase contains crucial growth factors and is in principle non-immunogenic and non-tumorigenic, thereby avoiding two major safety hurdles in the application of stem cells in regenerative medicine.

## 5. Conclusions

In conclusion, our analysis indicates that IStemRewind, with a minimal intervention by the operator, allows us to isolate from lipoaspirates ADSCs that are qualitatively comparable to ASDCs isolated using standard enzymatic techniques. Moreover, IStemRewind isolates contain immunomodulatory molecules and are devoid of proinflammatory cytokines.

## 6. Patents

GenLife is the owner of the patented method (SM-P-201700431, IT201700102994A1, PCT/IB2018/001077) through the patented medical device IStemRewind (SM-P-202000252, IT102020000010879, PCT/IB2021/054097).

## Figures and Tables

**Figure 1 biomedicines-11-01006-f001:**
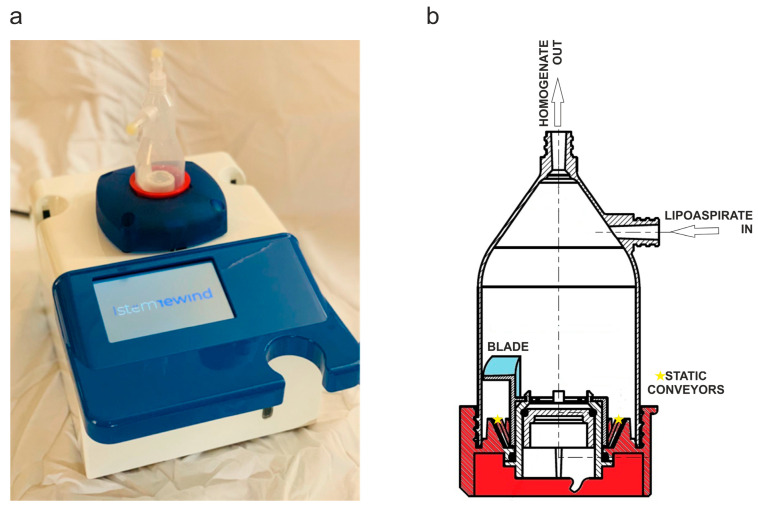
The IStemRewind device and the tube. (**a**) A picture of the IStemRewind system, a biological extractor equipped with a sterile disposable kit (IStemRewind tube). (**b**) A scheme of the IStemRewind tube structure: the entrance port of the biological sample (PORT A); the blade; the static conveyors and the door to extract the homogenate after the processing (PORT B) are indicated.

**Figure 2 biomedicines-11-01006-f002:**
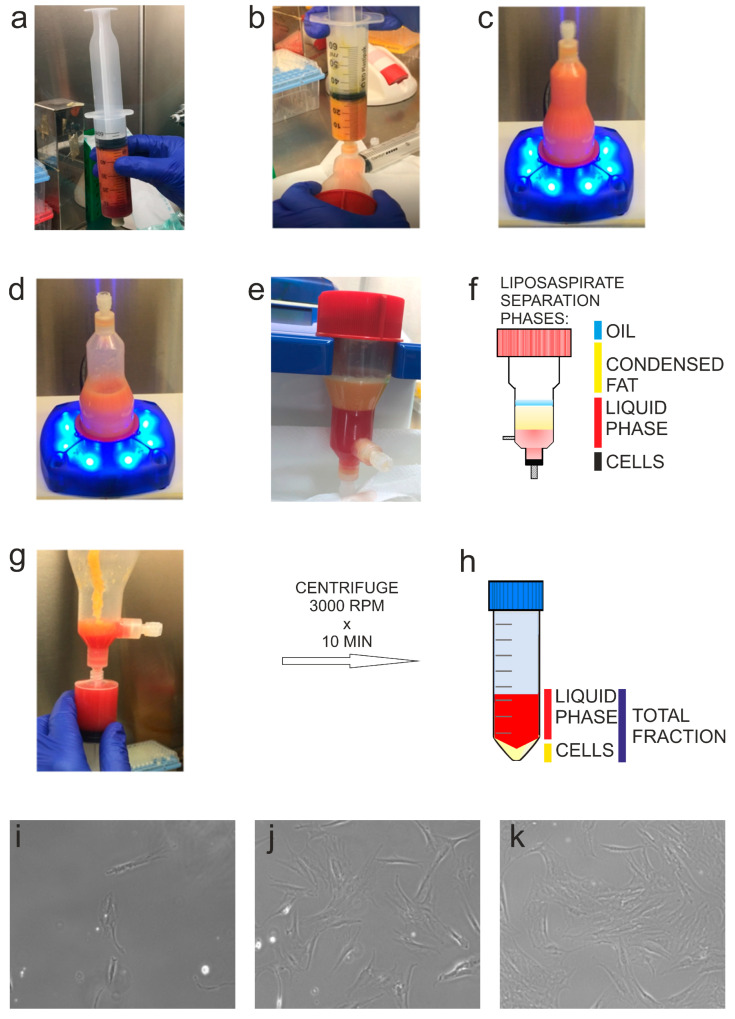
Lipoaspirate processing using IStemRewind. (**a**–**g**) Representative photographs of the different steps of the isolation procedure. The lipoaspirate sample, contained in a 50 mL sterile syringe (**a**), was injected into the IStemRewind tube (**b**) together with 20 mL of sterile saline solution. The IStemRewind tube with the sample was connected to the device and processed at 3000 rpm for 50 s (**c**), held for 10 s (**d**) and further processed for 20 s at 600 rpm. The IStemRewind tube was then positioned upside down for 15 min (**e**) to separate the different phases. (**f**) Picture of the different fractions isolated using IStemRewind, from top to bottom: oil, condensed fat, liquid phase (LP) and cells. (**g**) The total fraction (TF) composed of the LP and the cells was extracted from the IStemRewind tube with a sterile syringe and transferred to a 50 mL tube. (**h**) Picture of the appearance of the tube containing the TF after centrifugation. From top to bottom: LP and cells. (**i**–**k**) Representative bright field images of cells from the IStemRewind isolates seeded on plastic and kept in DMEM for 4 (**i**), 8 (**j**) and 16 days (**k**). Magnification: 40×.

**Figure 3 biomedicines-11-01006-f003:**
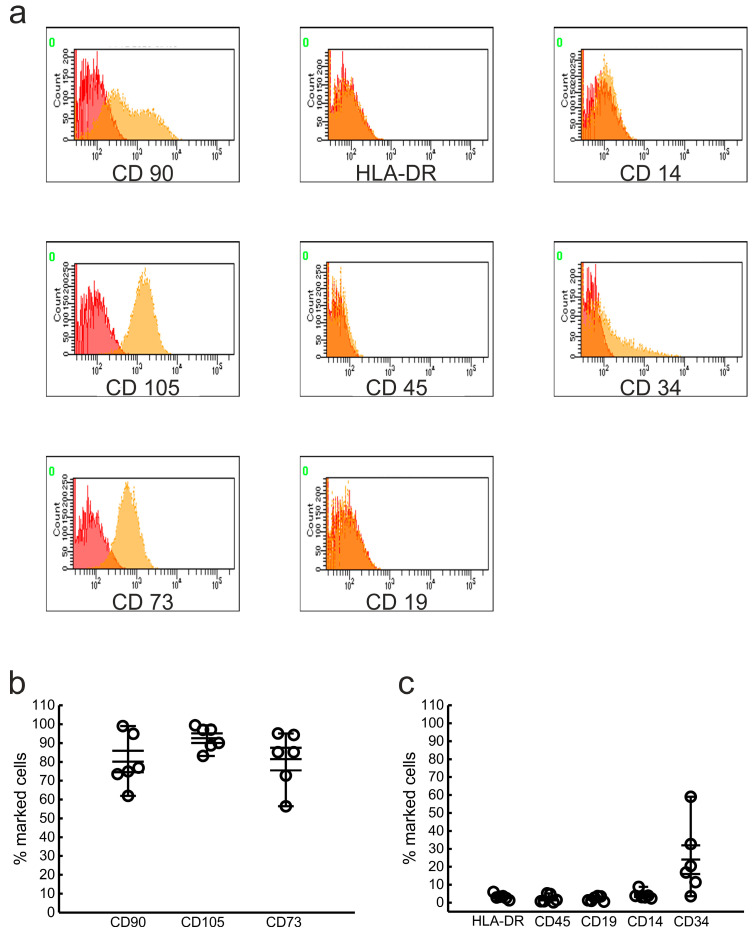
Immunophenotypic characterization of the cellular fraction isolated using IStemRewind from lipoaspirates. (**a**) Representative flow cytometry histograms of the indicated surface markers in cells from one IStemRewind isolate. Histograms indicate stained cells (yellow) over their respective isotypic control (red). (**b**,**c**) Box plots of the percentage of cells extracted from different lipoaspirates (*n* = 6) stained for mesenchymal markers (CD 90, CD 105 and CD 73, (**b**)) and hematopoietic markers (HLA-DR, CD 45, CD 19, CD 14, CD 34, (**c**)).

**Figure 4 biomedicines-11-01006-f004:**
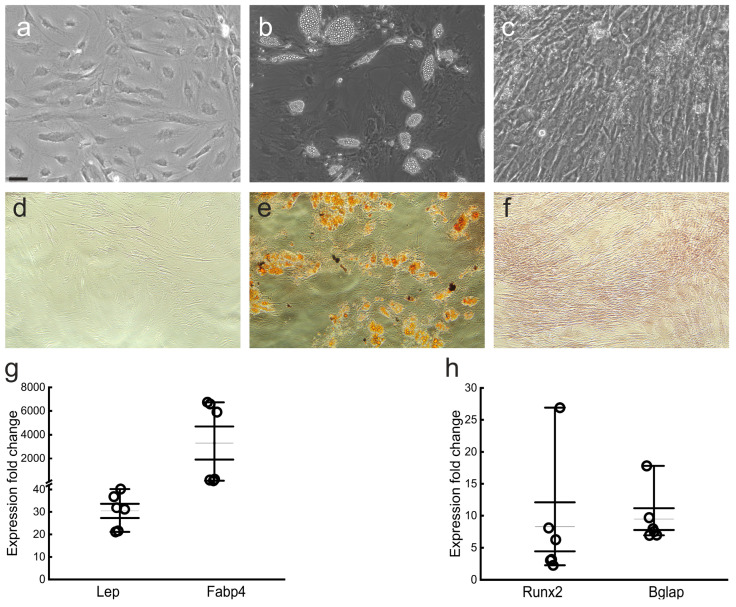
ADSCs isolated using IStemRewind can differentiate into adipocyte and osteocytes. (**a**–**f**) Representative bright field images of ADSCs isolated using IstemRewind. Cells were cultured in expansion medium (**a**,**d**) or in adipogenesis (**b**,**e**) and osteogenesis (**c**,**f**) inductive conditions. Adipocytes were stained with Oil Red O (sigma) and osteocytes were stained using Alzarin Red S (**f**) (magnification: 40×. Scale bar, 100 µm.). (**g**) Box plot of the expression of the adipocyte markers *Leptin* and *FABP4* measured by qRT-PCR in ADSCs after 4 days of adipogenic differentiation. (**h**) Box plot of the expression of the osteocyte markers *Osteocalcin* and *RUNX2* measured by qRT-PCR in ADSCs after 10 days of osteocyte differentiation. Values in g and h represent fold changes over the undifferentiated ADSCs.

**Figure 5 biomedicines-11-01006-f005:**
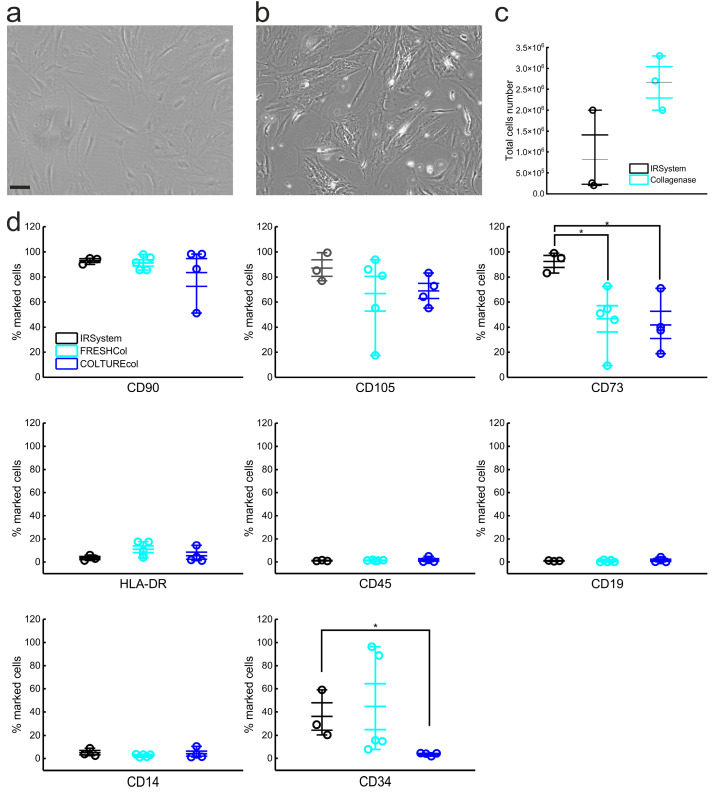
The cellular fraction isolated from lipoaspirates using IStemRewind contains CD73^+^ and CD34^+^ MSCs. (**a**,**b**) Representative bright field images of cells extracted with IStemRewind (**a**) or by collagenase digestion (**b**) from lipoaspirates. Scale bar, 100 µm. (**c**) Box plots of total cell counts determined by flow cytometry in cellular materials obtained from lipoaspirates processed as indicated (*n* = 3). (**d**) Box plots of the percentage of cells stained by the surface markers determined by flow cytometry in cells extracted using the indicated methods from lipoaspirates (*n* = 4). * *p* < 0.05 in a Kruskal–Wallis ANOVA among the indicated conditions.

**Figure 6 biomedicines-11-01006-f006:**
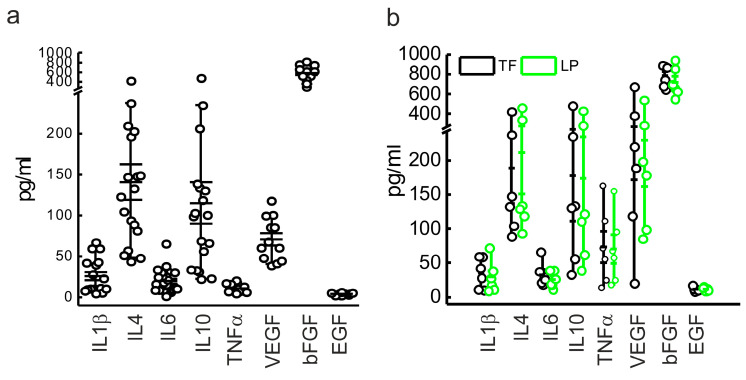
IStemRewind isolates contain more growth factors and immunomodulatory cytokines than pro-inflammatory cytokines. (**a**) Box plots of the concentration determined by specific ELISAs of the indicated proteins in TF extracted by using IStemRewind from 17 lipoaspirates. (**b**) Box plots of the concentration of the indicated proteins determined by specific ELISAs of the TF and in the LP purified by centrifugation from six randomly selected IStemRewind isolates.

**Figure 7 biomedicines-11-01006-f007:**
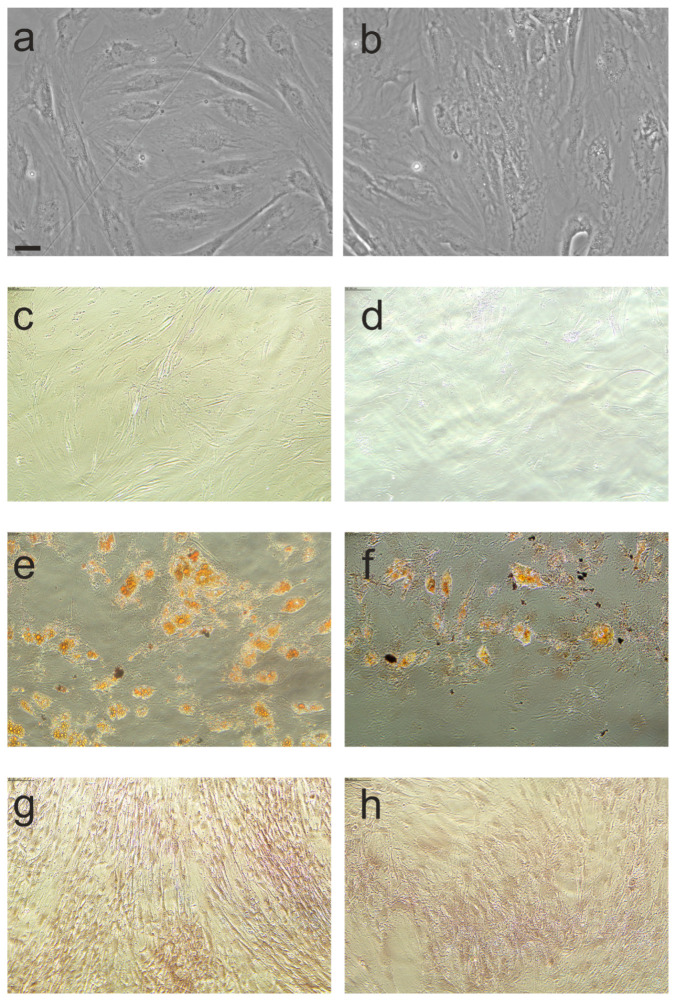
IStemRewind isolated ADSCs retain their differentiation potential after 1 month cryopreservation. (**a**–**f**) Representative bright field images of freshly isolated (**a**,**c**,**e**,**g**) and 1 month cryopreserved, thawed (**b**,**d**,**f**,**h**) MSCs isolated using IStemRewind. Scale bar, 100 µm. Cells were cultured in expansion medium (**a**–**d**), or in adipogenesis (**e**,**f**) and osteogenesis (**g**,**h**) inductive conditions. Adipocytes were stained with Oil Red O (**e**,**f**) and osteoblasts were stained using Alzarin Red S (**g**,**h**).

## Data Availability

The data presented in this study are available on request from the corresponding author.

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
