# Peer review of "A Novel Benchtop Device for Efficient and Simple Purification of Cytokines, Growth Factors and Stem Cells from Adipose Tissue"

_biomedicines, 2023, doi:10.3390/biomedicines11041006_

Round 1

Reviewer 1 Report

The authors described a novel benchtop device for cell purification without manipulation. They analysed the mesenchymal markers expression and differentiation capability. They also quantified some markers on the soluble fraction. The authors should well define the number of lipoaspirate samples between 17 or 18 and evaluate the viability and differentiation potential of isolated ADSCs even longer storage times.

The work could have a significant contribution to the regenerative medicine field.

Author Response

Please see the attached word doc

Reviewer 2 Report

In the manuscript entitled “A novel benchtop device for efficient and simple purification of cytokines, growth factors and stem cells from adipose tissue” the authors present a method for the enzyme-free isolation of adipose tissue-derived stem cells from lipoaspirates. ADSCs will become increasingly important in regenerative medicine and therefore the topic is of great interest and importance. The study is mostly well conducted and the results are presented conclusively. However, some concerns have to be addressed.

Major concern

The main problem with the manuscript is that the authors overestimate the relevance of their flow cytometry and ELISA data. It is nice that the cells can be isolated without enzymatic digestion and are positive for MSC-markers and have adipogenic and osteogenic differentiation capacity. However, the conclusions drawn from the ELISA and FACS data are not justified. For example: it is well known that CD34 is expressed in freshly isolated ADSCs and that the cells lose the positivity during cell culturing. A higher expression in comparison to the cultured cells is no surprise at all. Regarding the CD73 enrichment: ADSCs (or MSCs generally) are normally nearly 100% positive for CD73, CD90 AND CD105, meaning that every cell is positive for all three markers. How do the authors interpret this CD73-enrichment? Do the authors propose the existence of CD73+CD90-CD105- cells in the IStemRewind isolate responsible for the specific CD73-enrichment? Or are there CD73-CD90+CD105+ cells in the digestion isolate? I don’t think the CD73-enrichment concluded from the data is particular meaningful. And in regard to the ELISA data: what do the authors mean when they say there is an enrichment of the immunomodulatory cytokines IL4, IL10, bFGF and VEGF compared to the pro-inflammatory cytokines TNFα, IL1β and IL6? Enrichment in comparison to what? The absolute concentrations are no basis to conclude an enrichment. There are higher absolute concentrations of immunomodulatory cytokines and lower absolute concentrations of inflammatory cytokines. That is not an enrichment!

Don’t get me wrong, the study has nice results, and it is great that the cells can be isolated in this way. However, in my opinion the results from the FACS- und ELISA-data should not be used to declare some kind of superiority of the IStemRewind isolated cells compared with the cells from enzymatic digestion.

Minor concerns

2.1. Materials and reagents: did You really utilize a 5 % Pen/Strep-Solution? A 1 % solution is the concentration normally used.

2.7. Flow cytometry (and 2.9): did You really utilize cells in passage 9/10 for characterization? This is highly unusual for primary cells and maybe the experiment should be repeated for cells with no more than 4 or 5 passages.

3.1. Lipoaspirate Processing using IStemRewind: At the beginning of the paragraph too much method is reproduced (it should start with “After the sedimentation…”)

Figure 2, Figure legend: I don’t think “cartoon” is the best term, here.

The language needs a careful revision. Although it is always clear what the authors want to say, the syntax is somewhat confusing.

In the 6th line of the discussion: Do You mean CD34 or CD36?
